# Solvent-driven electron trapping and mass transport in reduced graphites to access perfect graphene

Philipp Vecera[1], Johannes Holzwarth[1], Konstantin F. Edelthalhammer[1], Udo Mundloch[1], Herwig Peterlik[2], Frank Hauke[1] & Andreas Hirsch[1]

Herein, we report on a significant discovery, namely, the quantitative discharging of reduced graphite forms, such as graphite intercalation compounds, graphenide dispersions and graphenides deposited on surfaces with the simple solvent benzonitrile. Because of its comparatively low reduction potential, benzonitrile is reduced during this process to the radical anion, which exhibits a red colour and serves as a reporter molecule for the quantitative determination of negative charges on the carbon sheets. Moreover, this discovery reveals a very fundamental physical–chemical phenomenon, namely a quantitative solvent reduction induced and electrostatically driven mass transport of $K^+$ ions from the graphite intercalation compounds into the liquid. The simple treatment of dispersed graphenides suspended on silica substrates with benzonitrile leads to the clean conversion to graphene. This unprecedented procedure represents a rather mild, scalable and inexpensive method for graphene production surpassing previous wet-chemical approaches.

[1] Department of Chemistry and Pharmacy and Joint Institute of Advanced Materials and Processes (ZMP), University of Erlangen-Nürnberg, Henkestrasse 42, Erlangen 91054, Germany. [2] University of Vienna, Faculty of Physics, Boltzmanngasse 5, Wien 1090, Austria. Correspondence and requests for materials should be addressed to A.H. (email: andreas.hirsch@fau.de).

Graphite intercalation compounds (GICs) involving alkali metals as interlayer guests[1–3] represent a well-established class of compounds and have found practical applications, for example, as reducing agents (KC8) in organic synthesis[4–6] or as lithium-ion batteries[7,8]. More recently, we[9–13] and others[14–16] have demonstrated that GICs are very suitable precursors for the covalent functionalization of graphene. In typical covalent functionalization sequences, the negatively charged graphene layers first act as reductants for electrophiles such as diazonium[17] or iodonium[12] compounds, alkyl iodides or protons[18], which are subsequently attacked by the intermediately generated organic radicals or H-atoms to give arylated[11], alkylated or hydrogenated[10] graphene. This wet-chemical functionalization concept is facilitated by the fact that due to Coulomb repulsion the negatively charged graphenide layers within the solid GICs can be dispersed in suitable organic solvents[13,15,17]. However, in this respect two fundamental issues have been overlooked so far: The first question is whether all negative charges of the graphenide intermediates can be controlled or even completely removed in such redox reactions[19]. Only the complete oxidation is expected to avoid reactions with moisture and oxygen during workup leading to side products with undesired and additional oxygen- and hydrogen functionalities. We have already reported on such side reactions in the field of reductive carbon nanotube chemistry[20] and will show below that these take indeed place also with the corresponding graphenides. The second goal is if the wet-chemical exfoliation of GICs into dispersed graphenide sheets can be used for the bulk production of defect-free graphene. Also this latter question is directly associated with the possibility of a controlled removal of all negative charges from the intrinsic air-sensitive graphenide intermediates. To the best of our knowledge, there are neither scalable methods nor any simple liquid oxidizing agents reported in literature, which are able to quantitatively discharge any type of graphenide solutions without a simultaneous alteration of the carbon framework.

We report here on a significant discovery, which gives answer to these questions and solves the associated challenges. Moreover, this discovery reveals a very fundamental physical–chemical phenomenon, namely a quantitative solvent reduction induced and electrostatically driven mass transport of K$^+$ ions from the GIC into the liquid. The fundamental finding is that the treatment of GICs with benzonitrile (PhCN), leads to a quantitative discharging of the individual graphenide sheets upon the formation of the coloured radical anion PhCN$^{\cdot-}$, which is easy to monitor quantitatively, the accompanying exhaustive and Coulomb force-driven migration of the potassium counter-ions from GICs into the surrounding benzonitrile phase, the suppression of any reactions of dispersed graphenides with moisture and air that is shown to take place when no treatment with benzonitrile is provided and the successful generation of defect-free single-layer graphene suspended on silica substrates. This latter discovery represents a rather mild, scalable, and inexpensive method for the wet-chemical graphene production.

## Results

**Solvent-driven oxidation of GICs quantified by UV/Vis spectroscopy.** Benzonitrile is mainly used as highly polar organic solvent, while the interaction with potassium GICs especially in the field of graphene research has not been fully evaluated yet. It has a comparatively low reduction potential of −2.74 V versus Ag in N,N-dimethylformamide (DMF). Generally, typical strong oxidizing agents show much higher reduction potentials, for instance tetracyanoquinodimethane with −0.2 V versus Ag in DMF[21–23]. Electropositive metals such as elemental potassium can be dissolved in PhCN, which is accompanied by the

formation of K$^+$ cations and PhCN$^{\cdot-}$ radical anions[24–26]. The red solution of PhCN$^{\cdot-}$ displays a broad absorption pattern with a peak at ∼390 nm and another feature of bands at 500 nm. For the determination of the extinction coefficient of PhCN$^{\cdot-}$, which has not been reported in literature so far, we performed a dilution series of potassium in benzonitrile. The linear correlation between amount of potassium and measured absorption reveals quantitative reactions (Fig. 1b). According to Lambert-Beer, the extinction coefficient of PhCN$^{\cdot-}$ was determined to be $\varepsilon_{390} = 4,000$ (±50) L mol$^{-1}$ cm$^{-1}$ (Supplementary Figure 1). This value is in the regime of literature estimates, but has never been exactly determined yet[24]. With this information at hand we set out to investigate the interaction between PhCN and GICs involving potassium as guest ions. For this purpose, we conducted a precise dilution series of pure potassium and various stages of

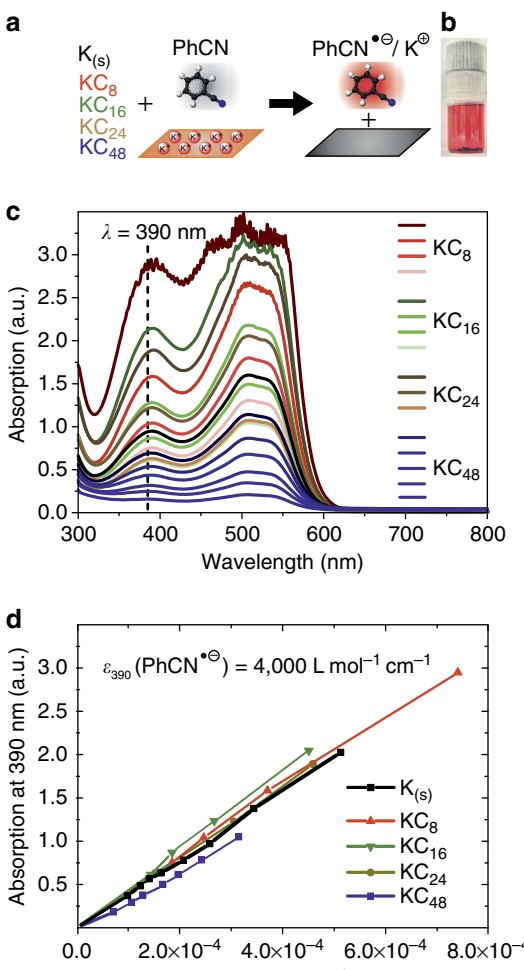

**Figure 1 | Charge quantification in GICs using UV/Vis spectroscopy.** (**a**) Reaction scheme for the quantitative electron transfer from various GICs to PhCN leading to dissolved K$^+$ ions and the red coloured radical anion PhCN$^{\cdot-}$. (**b**) Photograph of a sealed vial containing KC8 in a concentration of $5.0 \times 10^{-4}$ M in PhCN$^{\cdot-}$. (**c**) Absorption spectra under inert conditions of the dilution series of four different stages of GICs dispersed in PhCN. The resulting absorption profiles correlate to the benzonitrile radical anion PhCN$^{\cdot-}$. The dashed line indicates the wavelength used for the determination of $\varepsilon_{390}$. The exact concentration values—colour coded—are given in Supplementary Table 1. (**d**) Determination of the extinction coefficient by correlation of the potassium concentration versus the extinction at 390 nm. The black curve represents the dilution series of pure potassium K$_{(s)}$ in PhCN.

GICs with different K:C ratios, namely 1:0 (pure K), 1:8, 1:16, 1:24 and 1:48. Therefore, 480 mg (40 mmol carbon) spherical graphite SGN18 and 195 mg (5 mmol) potassium were heated to 200 °C in a glass vial. The formation of the final stage I intercalation compound was verified by *in situ* Raman spectroscopy under inert conditions (Fig. 2) as well as X-ray diffraction (XRD) analysis (Supplementary Figure 2). After the complete formation of the first stage K GIC, the powder was allowed to cool to ambient temperature. For the synthesis of $KC_8$ in PhCN (Fig. 1), the intercalation compounds were dissolved in absolute benzonitrile (0.15 mg ml$^{-1}$) by brief ultrasonication (5 min, 20 kJ, 1 s pulse) under argon atmosphere. The resulting formation of PhCN$^{\cdot-}$ was investigated by UV/Vis absorption spectroscopy in sealed cuvettes to exclude any air exposure (Fig. 1c,d).

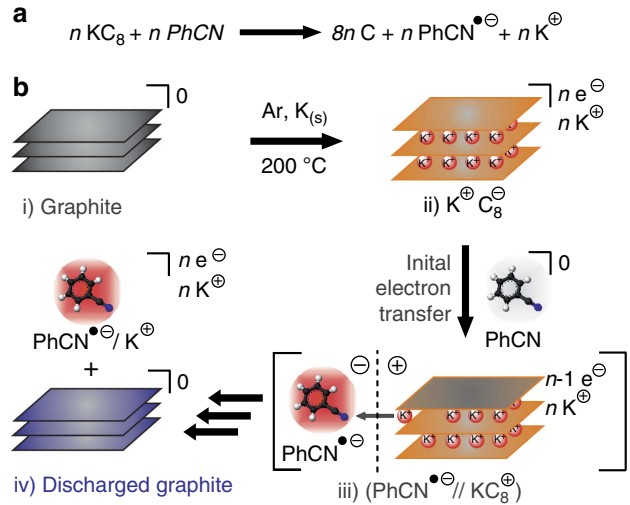

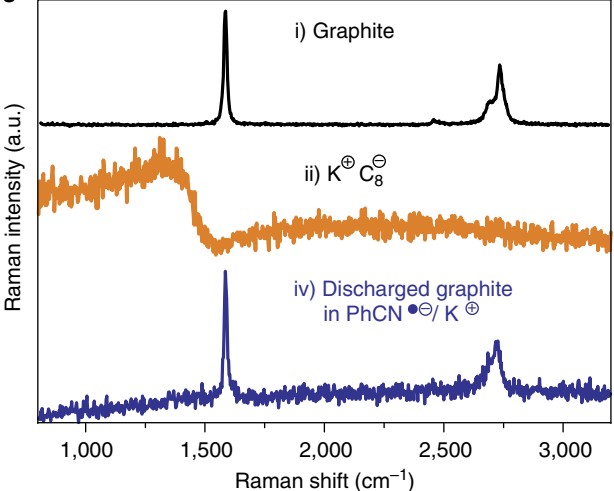

**Figure 2 | Charging and discharging of graphite.** (**a**) Reaction equation for the reaction of $KC_8$ with benzonitrile under inert gas atmosphere. (**b**) Reaction scheme: After graphite (i) and solid potassium are melted to (ii) $KC_8$, the addition of benzonitrile (PhCN) leads to an initial electron transfer from the GIC onto the PhCN. The so formed PhCN$^{\cdot-}$ anion and the resulting Coulomb attraction to the initially oxidized (iii) $KC_8^+$ is followed by deintercalation of $K^+$. After subsequent oxidation and $K^+$ migration, the (iii) fully discharged graphite is obtained while all charges are quantitatively transferred into the benzonitrile solution phase. (**c**) Raman spectra ($\lambda_{exc} = 532$ nm) of the respective species in the GIC/benzonitrile cell being stable under inert gas for months.

For the graphenide solutions of all GIC-stages, we determined the same linear concentration dependence as for pure K (Fig. 1d). These findings imply that all negative charges from the GICs are completely transferred to the benzonitrile and that residual charges on a GIC or on dispersed graphenide sheets can be determined based on the measured absorption of the corresponding PhCN$^{\cdot-}$ solutions.

**Charge and mass transport from GICs into benzonitrile solution.** An important question that arose from these investigations was what happens with the GICs after the quantitative discharging process. For addressing this point we applied Raman spectroscopy. To suitably handle the highly reactive intercalation compounds we carried out *in situ* Raman and X-ray diffraction (XRD) measurements under inert gas atmosphere thus inhibiting any reoxidation processes that could take place under ambient conditions. In Fig. 2, the entire potassium-promoted charging and PhCN-promoted discharging process of graphite is depicted.

After melting of potassium in the presence of graphite, the formation of a stage I GIC corresponding to $KC_8$ is verified by the characteristic broad Fano-line-shape spectrum of $KC_8$ (Fig. 2c)[27,28]. Besides the absence of any defect mode $\sim 1,330$ cm$^{-1}$, the G-mode is centred at 1,582 cm$^{-1}$ and therefore no residual doping is affecting the layers. The Raman 2D-mode at 2,700 cm$^{-1}$ provides the clear proof for restacked AB crystallinity in the crystal[29–32]. These Raman results are nicely corroborated by the XRD patterns, which also clearly prove the complete recovery (quantitative reoxidation and removal of intercalated $K^+$) of graphite after the treatment of the GIC with benzonitrile (Supplementary Figure 2). The XRD patterns of both $KC_8$ and recovered graphite with the interlayer distances of 3.35 Å (graphite) and 5.35 Å ($KC_8$) are exactly matching the literature values.

Obviously, the reoxidation of the negatively charged graphenide sheets and the formation of PhCN$^{\cdot-}$ anions is accompanied by mass transport of interlayer $K^+$ counterions from the interior of the solid GIC to the benzonitrile solution. This can be explained by an initial heterogeneous electron transfer reaction from the solid GIC to the liquid benzonitrile phase. As a consequence for a possible mechanism, it can be assumed that electrostatic forces are build up between the liquid phase, enriched with proximate, negatively charged PhCN$^{\cdot-}$ anions and the solid graphitic phase, enriched with positive charges (still intercalated $K^+$ counterions). At this step, the solvent-driven electron trapping induces an intermediately built-up Coulomb attraction between the liquid and solid phase, while the subsequent migration of the $K^+$ leads to energy minimization of the entire system which is indicated by the intermediate step *iii*) in Fig. 2. A related mass transport process takes place in the discharging process of Li-ion batteries in the solid state, but to the best of our knowledge has never been observed in simple binary systems consisting of a GIC and a redox-active solvent only[33]. We are aware of the fact that our discovery of the straightforward charge monitoring using PhCN$^-$ as a reporter molecule could find beneficial applications in battery technology. The current method for the state of charge determination by measuring voltage is frequently associated with errors for some materials.

For further product characterization we applied thermogravimetric analysis coupled to a mass spectrometer (TG-MS) of the generated bulk solid sample after workup under ambient conditions (Supplementary Figure 3a). The reoxidized graphitic material shows almost no weight loss ($\sim 2\%$) after heating to 550 °C under nitrogen atmosphere. This also demonstrates the full recovery of graphite after the treatment of the GIC with PhCN and corroborates the clean and complete discharging of

the graphene sheets. Statistical Raman spectroscopy (SRS)[34] analysis of the sample after workup fully proves the comprehensive conversion to graphite as can be seen by the counted Raman $I_D/I_G$ ratios (Supplementary Figure 3b) that match those in the histogram of pristine graphite (Supplementary Figure 6). Only a very small increase in D-mode intensity was detected, which is caused by traces of residual water and oxygen in the benzonitrile which presumably leads to some minor hydrogenation and oxygenation[35]. This minor defects however can be healed by thermally annealing as demonstrated in the Supplementary Figs 3c–d, since the D-mode vanishes at temperatures >400 °C which can be attributed to the cleavage of $sp^3$ bound moieties[36].

**Inhibition of covalent side-reactions via discharging in solution**. To systematically analyse the difference of the chemical reactivity of GICs and graphenide dispersions with and without prior PhCN treatment we carried out a series of comparison studies. For this purpose 24.0 mg of $KC_8$ were dispersed in 200 ml absolute tetrahydrofuran (THF) by tip ultrasonication (15 min, 20 kJ, 1 s pulse) as depicted in Fig. 3a. Then we divided the graphenide solution into two parts $G_A$ and $G_B$ of 100 ml, each. In contrast to $G_A$, the fraction $G_B$ was treated with an excess of 10.0 ml PhCN resulting in a colour change to red, which is indicative for the formation of the respective PhCN·⁻ anion in THF. Both fractions were subjected to aqueous workup under ambient conditions. The dispersions were filtrated and the resulting black powders were dried at 70 °C before the final characterization by TG-MS and SRS (Fig. 3). For comparison also pristine graphite $G_P$ was investigated under the same conditions.

Significantly, sample $G_A$ reveals a fundamental increase in the Raman $I_D/I_G$ intensity ratio, indicative for covalent adduct

formation (Supplementary Figure 4). As a direct result of the treatment with PhCN, the Raman D-mode for $G_B$ is only increased to a very small extent in comparison with $G_A$—almost the same $I_D/I_G$ intensity ratio is observed for the pristine graphite $G_P$ (Fig. 3b). Obviously, the side reactions leading to the formation of $G_A$ are completely suppressed. The data analysis of the TG-MS investigations (Fig. 3c,d) provides the final proof of the chemical identity of the defects generated during the formation of $G_A$ and clearly corroborates the Raman data, since $G_A$ experiences a mass loss of 13% in contrast to $G_B$ (4%). Mass spectrometry analysis of the cleavage products reveals fragments with $m/z$ 2 and 18 demonstrating the detachment of -H and -OH moieties in $G_A$. On the other hand, for $G_B$ these characteristic mass traces are almost absent and only a very small signal for $m/z$ 18 can be detected. These experiments provide the first experimental proof of the expected reaction of graphenides with water and oxygen—previously monitored for carbon nanotubes[20]. At the same time they corroborate the findings from above, namely, that graphenides can be completely reoxidized to neutral and chemically inert graphite.

**Production of graphene by graphenide reoxidation on substrates**. A direct consequence of these studies is that wet-chemical graphene production should be possible on the basis of this novel protocol, as long as reaggregation of the individualized graphene sheets to graphite can be suppressed. To prove this we have prepared graphenide solutions by dissolving 12.0 mg $KC_8$ in 100 ml dimethyl sulfoxide (DMSO). After 2 h of sedimentation, monolayer graphene films (1LG) on Si/SiO₂ wafers were formed via drop casting of one drop in the glovebox.

These surface depositions prevent subsequent reaggregation and stacking of the individual layers. The graphenide deposits

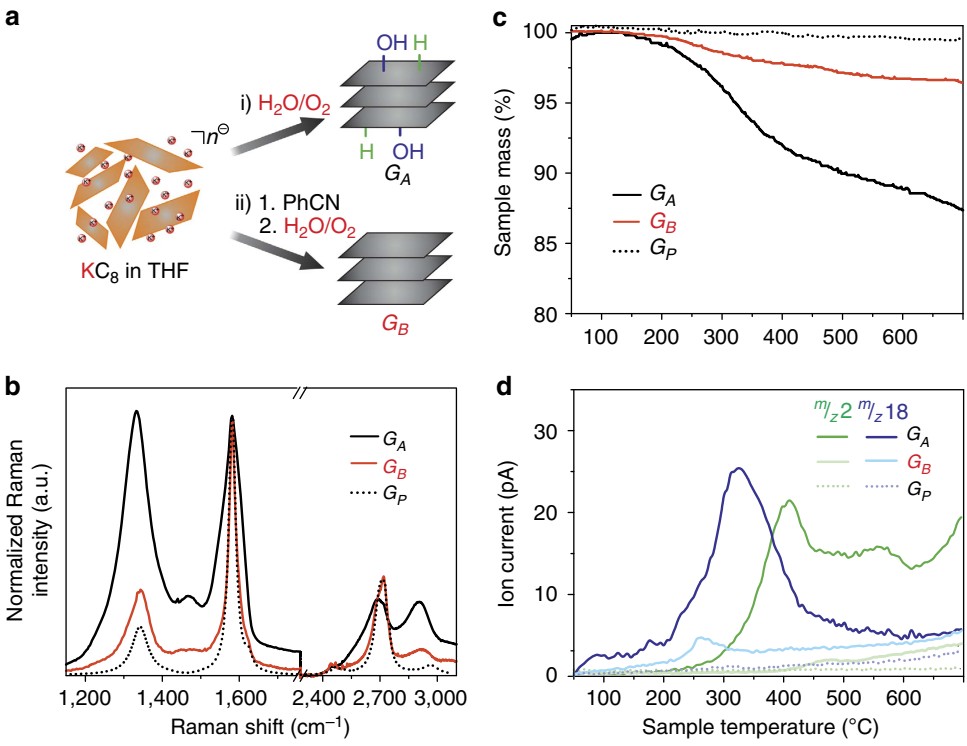

**Figure 3 | Oxidative side-reactions of graphenides and the inhibition by PhCN treatment.** (**a**) The sample is divided into two parts ($G_A$ and $G_B$). In contrast to $G_A$, $G_B$ was treated with PhCN before exposure to ambient conditions. The dotted line indicates pristine graphite ($G_P$) being dispersed in THF and PhCN as reference. (**b**) Statistical Raman spectroscopy mean spectra of 10,000 measured pixels (100 × 100 µm). (**c**) TG mass loss profiles of $G_A$, $G_B$ and the reference $G_P$. (**d**) Correlating MS traces for $m/z$ 2 and 18 indicating hydrogenation and hydroxylation for $G_A$ and vanishing ion current detection in the case of the PhCN treated sample $G_B$ compared with the reference $G_P$.

were subsequently either directly exposed to ambient conditions ($1LG_{(ac)}$) or subjected to treatment with PhCN ($1LG_{(PhCN)}$). The so obtained graphene flakes exhibit single-layer nature bearing a mean height of 1.5 nm over a lateral size of 10–18 μm grain size in pristine graphite SGN18—as depicted in the atomic force microscopy (AFM) image in Fig. 4b. It has to be noted that adsorbates on graphene and sheet overlapping lead to an overestimation of the real height in the AFM tapping mode[37]. By accounting ∼1 nm for one layer graphene, the verification of single-layer graphene has to be carried out via a direct correlation using scanning Raman microscopy and is reflected in the intensity map in Fig. 4c. Here, the overlapping of two single layers (0.8 nm height in the AFM, 300 counts for the Raman G-mode) can be regarded in the centre of the AFM image. For a broad and significant analysis of multiple flakes, we characterized a large area film of $1LG_{(PhCN)}$ by SRS (Fig. 5a–c).

The samples $1LG_{(ac)}$ display a mean $I_D/I_G$ ratio of 2.5 which correlates with an average degree of defects of $\theta = 0.5\%$. This shows that also on a substrate the deposited graphenides undergo chemical reactions leading to $sp^3$-defects. In the case of $1LG_{(PhCN)}$, however, the characteristic Raman features of high-quality single-layer graphene are observed. In particular, the D-mode intensity is almost vanishing, indicative for a very low density of $sp^3$ carbon atoms ($\theta = 0.003\%$). Moreover, the 2D-mode exhibits the typical narrow Lorentzian line shape and a drastically increased intensity with respect to the G-mode. In fact the spectroscopic features of $1LG_{(PhCN)}$ reveal the formation of defect-free graphene, with even superior quality with respect to chemical vapour deposition grown samples[38]. This high quality of

our wet-chemically prepared graphene is also demonstrated in the respective SRS scatterplot displayed in Fig. 5c, where the $I_{2D}/I_G$ ratio in relation to the $I_D/I_G$ ratio is presented for each data point within the investigated area of the Raman map. The sample quality of the graphene prepared in this way is extraordinary and can easily compete with bottom-up synthetic routes[39].

## Discussion

The discovery of a facile and quantitative electron trapping from reduced graphite forms such as graphite interaction compounds (GICs), graphenide dispersions, and graphenides deposited on surfaces by simple exposure to benzonitrile opens unprecedented perspectives in graphene chemistry and engineering. Benzonitrile is a common and cheap organic solvent, which at the same time has a rather low reduction potential. This allows for a quantitative discharging of reduced graphites in a heterogeneous solid/liquid phase reaction—a quantitative process that has been overlooked so far. At the same time the red coloured benzonitrile radical anion ($PhCN^{\cdot-}$) serves as an ideal reporter molecule for the quantification of the amount of negative charges on GICs or graphenides using absorption spectroscopy. Associated with the discharging of the GICs and the documented conversion to the parent graphite is the quantitative migration of the interlayer potassium cations into the liquid phase. The mass transport is driven by the electrostatic potential caused by adlayers of initially formed $PhCN^{\cdot-}$ anions on the surface of the GIC and their diffusion into the liquid phase. The opportunities of this unprecedented and simple solvent-driven mass transport

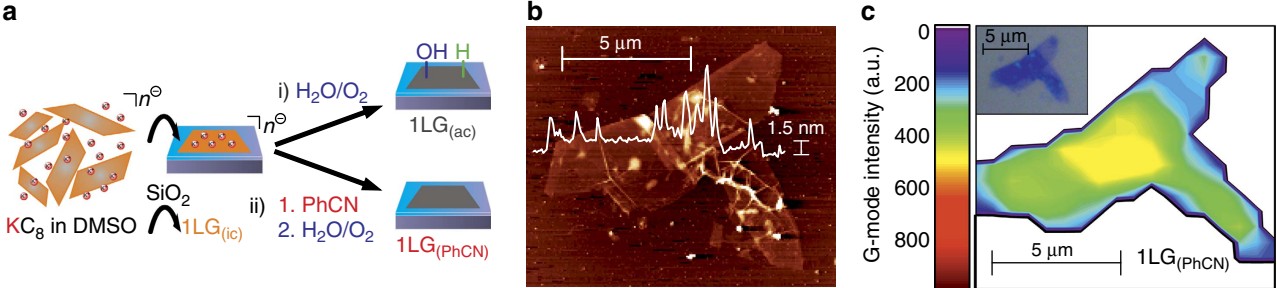

**Figure 4 | Production of large-area graphene films for SRM analysis.** (**a**) Dispersions of $KC_8$ in DMSO and drop casting of the corresponding graphenide solutions onto Si/$SiO_2$ wafers ($SiO_2$ layer thickness of 300 nm) under inert conditions leading to air-sensitive graphenide flakes $1LG_{(ic)}$. Before analysis, $1LG_{(ic)}$ was (i) either directly exposed to ambient conditions ($1LG_{(ac)}$, black) or (ii) previously washed by PhCN ($1LG_{(PhCN)}$, red). (**b**) AFM image (scale bar 5 μm) with height profile indicating the single-layer nature of the obtained $1LG_{(PhCN)}$ with lateral dimensions of ∼10 μm with a height of ∼1.5 nm. Note that the overlap of two single layers in the centre of the flake can be revealed by the underlying z-profile. (**c**) Corresponding SRM map showing the respective G-mode intensity in relation to the optical image shown in the inset.

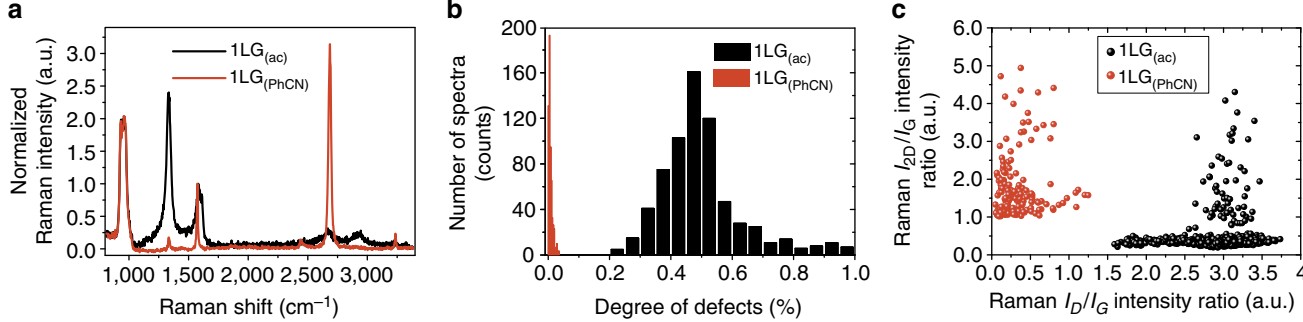

**Figure 5 | Statistical Raman spectroscopy of exfoliated graphene.** (**a**) Comparison of representative Raman spectra carried out on large-area films ($50 \times 50$ μm$^2$). (**b**) Statistical histogram representation of defect densities of the entire SRM map in Fig. 4c. (**c**) Scatterplot of the Raman intensity ratio dependency: The $I_D/I_G$ versus $I_{2D}/I_G$ Raman intensity ratio reveals the effect of PhCN retaining the high quality of $1LG_{(PhCN)}$ in contrast to air exposed $1LG_{(ac)}$ graphene samples.

(which is related to that in lithium-ion batteries) for applications in energy release technologies did not escape our attention. We have also demonstrated direct practical use of our discovery, namely, the straightforward formation of graphene from graphenides deposited on surfaces. The advantage of this simple and inexpensive graphene production surpasses previous wet-chemical methods such as exfoliation of neutral graphite in solvents and the reduction of graphene oxide, since both features are combined in our method namely, the complete exfoliation down to single-layer sheets and the application of rather mild reaction conditions avoiding a priori any formation of $\sigma$-defects (holes) in the carbon plane.

## Methods

**Raman spectroscopy.** Raman spectroscopic characterization was carried out on a Horiba Jobin Yvon LabRAM Aramis confocal Raman microscope (excitation wavelength: 532 nm) with a laser spot size of ~1 μm (Olympus LMPlanFl 50x, NA 0.50). Raman measurements were carried out using a micro-Raman setup in backscattering geometry. A charge-coupled device is used to detect the signal after analysing the signal via a monochromator. The spectrometer was calibrated in frequency using a HOPG crystal. Statistical Raman measurements were obtained through a motorized x–y table in a continuous linescan mode (SWIFT-module) in order to obtain faster measurements. In situ Raman measurements were carried out in a quartz tube sealed by an ISO-KF flange in order to keep inert conditions. The laser power was kept below 0.01 mW to avoid laser induced deintercalation of intermediate species.

**AFM.** Atomic force microscopy was carried out on a Solver Pro scanning probe microscope equipped with NSG-10 cantilevers (NT-MDT) and a Sony Exwave HAD camera optical zoom (6.5) in tapping mode.

**Thermogravimetric analysis and mass spectrometry (TG-MS).** For the analysis of $m/z > 45$, the thermogravimetric analysis of pure PhCN solutions was carried out on a Perkin Elmer Pyris 1 TGA instrument. For the analysis of $m/z < 45$ ($H_2$, $H_2O$) in $G_A$ and $G_B$ (Fig. 3), the TG-MS data was recorded on a Netzsch STA 409 CD instrument equipped with a Skimmer QMS 422 mass spectrometer (MS/EI). Details are provided in the Supplementary Method.

**Glovebox.** Sample preparation, solvent processing, functionalization was carried out in an argon-filled Labmaster SP glovebox (MBraun), equipped with a gas filter to remove solvents and an argon cooling systems, with an oxygen content <0.1 p.p.m. and a water content <0.1 p.p.m.

**X-ray diffraction.** X-ray diffraction was performed by placing the material in a glove box into glass capillaries with 1.5–2 mm diameter and 10 μm wall thickness (Hilgenberg, Germany) and subsequent sealing. X-ray patterns were measured using a microfocus X-ray source with a copper target ($\lambda = 1.542$ Å) equipped with a pinhole camera (Nanostar, Bruker AXS) and an image plate system (Fujifilm FLA 7,000). All two-dimensional X-ray images were radially averaged and background corrected to obtain the scattering intensities in dependence on the scattering angle $2\theta$.

**Graphite.** As starting material a spherical graphite SGN18 (Future Carbon, Germany), a synthetic graphite (99.99 %C, <0.01% ash) with a comparatively small mean grain size of 18 μm (Supplementary Figure 5), a high specific surface area of 6.2 $m^2 g^{-1}$ and a resistivity of 0.001 $\Omega$ cm was chosen. An average Raman $I_D/I_G$ intensity ratio of 0.3 is present in the starting material (Supplementary Figure 6).

**Potassium.** Potassium chunks were bought from Sigma-Aldrich Co. and used as received.

**Benzonitrile (PhCN).** Benzonitrile (PhCN) was received anhydrous from Sigma-Aldrich Co. and dried to absolute quality (<1 p.p.m. $H_2O$, <1 p.p.m. $O_2$) over molecular sieves (3 Å). Finally, pump–freeze technique was used to completely degas the solvent before the reaction.

**Tetrahydrofuran (THF).** THF was received anhydrous from Sigma-Aldrich Co. and dried over molecular sieves (3 Å). Subsequently, it was distilled over Na/K alloy to remove inhibitor and achieve absolute quality (<1 p.p.m. $H_2O$, <1 p.p.m. $O_2$). Finally, pump–freeze technique was used to completely degas the solvent prior to use.

**Dimethyl sulfoxide (DMSO).** DMSO was received anhydrous and purified (99.99%) from Sigma-Aldrich Co. and pump–freeze technique was used to completely degas the solvents before the reaction.

**Data availability.** The data supporting the findings of this study are available within the article and its Supplementary Information files are available from the corresponding author upon request.

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

## Acknowledgements

We thank the Deutsche Forschungsgemeinschaft (DFG - SFB 953 'Synthetic Carbon Allotropes', Project A1) and the Graduate School Molecular Science (GSMS) for financial support. The research leading to these results has received partial funding from the European Union Seventh Framework Programme under grant agreement no. 604391 Graphene Flagship.

## Author contributions

A.H. and F.H. supervised the project as scientific group leader and principal investigator. P.V. discovered the effect, worked out the concept, produced the samples and carried out Raman spectroscopy and TG-MS. K.F.E. prepared solvents to absolute conditions and performed ultraviolet/visible *in situ* experiments. J.H. carried out the electrochemical references, contributed to the concept and provided scientific input. U.M. recorded the optical and AFM image. H.P. performed the X-ray diffraction experiment and analysis. P.V. and A.H. wrote the manuscript.

## Additional information

**Competing financial interests:** The authors declare no competing financial interests.

