## [Peer review file · Nature Communications]

Reviewers' comments:

Reviewer #1 (Remarks to the Author):

This is interesting work that has been executed carefully. However, the complete discharging of the graphenide to benzonitrile would be expected based on the earlier work of Hirsch and others. Evidence is presented for a new synthesis of single layer graphene. The AFM image shown as figure 4b is difficult to interpret, but it is claimed that the height of the graphene is 1.5 nm. This could be as many as 4 layers of graphene. This would need to be clarified before publication in Nature Communications is warranted.

Reviewer #2 (Remarks to the Author):

The authors present an interesting finding that KCx can be quantitatively oxidized by benzonitrile. The potential use is mainly that this process does not result in covalent functionalization of the graphene sheets (vs water oxidation), and the reaction can be used for redox titration to indicate for example the extent of charge on graphene layers.

The first application the focus, with significant characterization indicating that the resulting graphene is relatively defect free compared with the reduced sheets exposed to water or air.

I think the reaction is not surprising, but the results could prove very useful to the graphene community. I had some major and some minor suggestions to the authors for revision.

major points

1. There is a fundamentally important issue of whether all the K metal has reacted with graphite to make KCx. It may well be there is some, or even a lot, of unreacted K remaining. The paper nicely shows that this would react similarly with benzonitrile. I don't think the Raman data are sufficient to make this point clear, as they would likely look indistinguishable for GICs with stage 1, 2 etc., or for different charging within a stage. At a minimum, I think we need powder XRD or other methods to be clear that the K metal all reacted.

2. There is one more comparison that would be critical in showing the importance of this technique, because the alternative of using water for the GIC re-oxidation is so simple. The question is, how does a water-oxidized, and then thermally-treated, sample compare to one oxidized with benzonitrile? If there is not much difference, then the former route might be simpler and preferred.

minor points

1. In a few places, including the title, the authors used some pretty awkward phrasing to indicate oxidation, like "electron capture" or "removal of negative charges". I would suggest replacing these with "oxidation"

2. on p.3, K are described as "guest atoms". Given the content of this article, this must be changed to "guest ions" and refer to K^+

3. on p.6, and elsewhere, the authors indicate this is the first redox-active solvent reaction. But the well-known reaction of KC_8 with water, or any number of other reductively-unstable solvents, could be included in this class, too.

4. on p.6, the text description on oxidation should probably be redone. The de-intercalation of K^+ from the graphene galleries can simply be understood from charge neutrality requirement when the graphite sheets are oxidized. No need to invoke solvent effects here.

Reviewer #3 (Remarks to the Author):

<Discovery of Solvent Driven Electron Trapping and Mass Transport in Reduced Graphites - Access to Perfect Graphene> suggests new reaction phenomenon in GIC; adopting PhCN (a solvent that can easily accommodate electron) induces mass transport of K^+ from GIC. Especially, PhCN⁻ anion shows red color, providing a quantitative measuring method of charge transfer. Finally, the authors applied this new method to the production of high-quality graphene. The new synthetic method is quite novel compared to the conventional graphene-synthesis method, and it seems to have possibilities of making breakthrough in GIC-related-synthesis. However, several parts should be revised as follows.

C1. The authors have mainly used K^+ as an intercalant and PhCN for a solvent, and I wonder why the authors have chosen those materials among various candidates. To generalize the reaction that is so-called "solvent-reduction-induced and electrostatically driven mass transport", synthesis with other various intercalants and solvents may be done. For example, K^+ can be replaced to Rb^+ , Cs^+ , or $[Na-ether]^+$ and PhCN might be changed to other electron-trapping solvents.

C2. In figure 2, the authors claim that K^+ diffuses out of graphite galleries by showing Raman spectroscopy data. However, I would like to ask X-ray diffraction result of K^+ -intercalated graphite (KC_8) and KC_8 after mixing with PhCN. The additional XRD data will clarify the structure of graphite (before and after exposure to PhCN) and will allow quantitative analysis of how much K^+ goes out of graphite (in other words, the initial and final stage number).

C3. In page 9 line 7, "To proof" should be changed into "To prove".

C4. In figure 4, it was shown that KC_8 treated with PhCN produces higher-quality graphene compared to the one which were not treated with PhCN. That is to say, "electron trapping from reduced graphite" stimulates the exfoliation of graphite even without any ultrasonication. How and why does it happen? Would you explain fundamental reaction mechanism in detail with supporting data?

C5. The coloring of reacting species seems to have novelty in battery applications. The conventional battery system estimates SOC (state of charge) by measuring voltage, but it makes errors for some materials. However, the suggested method is expected to provide a new way of measuring the amount of charge in graphite. Therefore, additional comments about this would provide broader aspects to the readers.

Author response to reviewer comments

Reviewer #1 (Remarks to the Author):

This is interesting work that has been executed carefully. However, the complete discharging of the graphenide to benzonitrile would be expected based on the earlier work of Hirsch and others. Evidence is presented for a new synthesis of single layer graphene. The AFM image shown as figure 4b is difficult to interpret, but it is claimed that the height of the graphene is 1.5 nm. This could be as many as 4 layers of graphene. This would need to be clarified before publication in Nature Communications is warranted.

We thank the referee for the kind response on our manuscript in relation to the earlier work in the field of carbon allotrope functionalization and the qualified comment about the flake structure presented in the AFM image (Figure 4b).

The measured thickness in the center of the graphene flake of up to 1.5 nm is higher than the expected single atomic layer of about 0.4-0.5 nm. This can be explained by adsorbates, partial folding and overlapping of single layers that can occur by wet-chemical deposition of graphene sheets on surfaces. Such phenomena have been observed before, for example by Nemes-Incze, P. *et al.* Anomalies in thickness measurements of graphene and few layer graphite crystals by tapping mode atomic force microscopy. *Carbon* **46**, 1435-1442 (2008). As a consequence, the flake heights are easily overestimated. However, by taking also into account the spectroscopic information obtained from SRM (Figure 4c-f), there is no doubt that decoupled *a priori* single layer graphene sheets have been generated and investigated. The ratio and shape of the 2D and G modes are a clear proof for the presence of single layer graphene. In the center of the flake, the deposition of the graphene sheets led to an overlapping of two individual graphene sheets. This can be revealed by the steps in the z-profile from the AFM image as well as the G-mode intensity in the SRM map (Fig. 4c) and has been addressed in the manuscript.

Reviewer #2 (Remarks to the Author):

The authors present an interesting finding that KCx can be quantitatively oxidized by benzonitrile. The potential use is mainly that this process does not result in covalent functionalization of the graphene sheets (vs water oxidation), and the reaction can be used for redox titration to indicate for example the extent of charge on graphene layers.

The first application the focus, with significant characterization indicating that the resulting graphene is relatively defect free compared with reduced sheets exposed to water or air. I think the reaction is not surprising, but the results could prove very useful to the graphene community. I had some major and some minor suggestions to the authors for revision.

We thank the reviewer for this very positive evaluation of our work.

major points

1. There is a fundamentally important issue of whether all the K metal has reacted with graphite to make KCx. It may well be there is some, or even a lot, of unreacted K remaining. The paper

nicely shows that this would react similarly with benzonitrile. I don't think the Raman data are sufficient to make this point clear, as they would likely look indistinguishable for GICs with stage 1, 2 etc., or for different charging within a stage. At a minimum, I think we need powder XRD or other methods to be clear that the K metal all reacted.

We fully agree with the reviewer that XRD is an independent measure for the intercalation or deintercalation of guest atoms into graphite. As a consequence, the determination of XRD patterns will greatly strengthen our statement which based on Raman spectroscopy. Therefore we performed the requested XRD measurements of the graphite used as starting material, the KC_8 performed under inert conditions and the benzonitrile treated KC_8 . We were very pleased to provide independent proof for the complete recovery (quantitative reoxidation and removal of intercalated K^+) of graphite after the treatment of the GIC with benzonitrile (Supplementary Fig. 2). The XRD patterns of both KC_8 and recovered graphite with the interlayer distances of 3.35 Å (graphite) and 5.35 Å (KC_8) are exactly matching the literature values. From the XRD data, we can also conclude that the entire K has reacted to KC_8 , while after PhCN treatment a full recovery of the graphitic reflections is observed. Thus, the XRD fully supports the Raman data in every respect. Moreover, these findings imply that in future, *in situ* Raman spectroscopy can be used as a self-contained tool for distinguishing of intercalated from parent graphite.

2. There is one more comparison that would be critical in showing the importance of this technique, because the alternative of using water for the GIC re-oxidation is so simple. The question is, how does a water-oxidized, and then thermally-treated, sample compare to one oxidized with benzonitrile? If there is not much difference, then the former route might be simpler and preferred.

The bulk sample analysis in Figure 3 proved that oxidation under ambient conditions leads to the covalent hydroxylation and hydrogenation of the graphene lattice. As we have demonstrated recently, the functionalization of graphenides with alkyl/aryl halides, diazonium- or iodonium salts leads to alkylated/arylated graphene monolayer derivatives (References 9, 11-13, 15,17), while the treatment with water can be used for the hydrogenation of graphene (Ref. 10, 18).

This demonstrates that water cannot be used for a simple oxidation, since also hydrogenation of graphene is a preferred reaction pathway. If one would like to produce intact graphene from hydrogenated graphene, a subsequent annealing step is required which makes the process very inappropriate compared to the straightforward inert oxidation with benzonitrile.

Moreover, as we have demonstrated in this manuscript, the exposure of graphenides to O_2/H_2O (ambient conditions), also leads to hydroxylation (Figure 3). Consequently, the discharging of exfoliated graphene by PhCN inhibits covalent hydrogenation and hydroxylation, thus leading to defect-free graphene (Figure 4). This PhCN treatment of graphenides represents the first synthetic route for the direct production of defect-free graphene from graphenide solutions in combination with direct quantification of the amount of charges.

minor points

1. In a few places, including the title, the authors used some pretty awkward phrasing to indicate

oxidation, like "electron capture" or "removal of negative charges". I would suggest replacing these with "oxidation"

We followed the advice of the reviewer and replaced the terms mentioned above in the manuscript by "oxidation".

2. on p.3, K are described as "guest atoms". Given the content of this article, this must be changed to "guest ions" and refer to K⁺

We changed the term as requested.

3. on p.6, and elsewhere, the authors indicate this is the first redox-active solvent reaction. But the well-known reaction of KC_x with water, or any number of other reductively-unstable solvents, could be included in this class, too.

As we have pointed out above, water reacts with graphenides also to covalent adducts (References 10, 18) which makes it inappropriate for the production of defect free graphene. However, recent reports about stabilization of charged graphenides in degassed water indicate a reoxidation with low degrees of defects in the basal plane (<http://arxiv.org/abs/1603.05421>). Nevertheless, no fully inert and stable organic solvent has been reported which is able to oxidise GICs without any functionalization.

4. on p.6, the text description on oxidation should probably be redone. The de-intercalation of K⁺ from the graphene galleries can simply be understood from charge neutrality requirement when the graphite sheets are oxidized. No need to invoke solvent effects here.

We agree with the referee that charge neutrality requirement accords for the macroscopic view on the bulk sample. However, we were able to provide detailed insights into the mechanism by *in situ* Raman spectroscopy and additional XRD analysis. Therefore, the Coulomb-driven mass transport from the solid state into the solvent was recognized and has not been addressed previously. This represents a fundamental process in the field of graphite intercalation chemistry and has been rationalized in our work for the first time.

Reviewer #3 (Remarks to the Author):

<Discovery of Solvent Driven Electron Trapping and Mass Transport in Reduced Graphites - Access to Perfect Graphene> suggests new reaction phenomenon in GIC; adopting PhCN (a solvent that can easily accommodate electron) induces mass transport of K⁺ from GIC. Especially, PhCN⁻ anion shows red color, providing a quantitative measuring method of charge transfer. Finally, the authors applied this new method to the production of high-quality graphene. The new synthetic method is quite novel compared to the conventional graphene-synthesis method, and it seems to have possibilities of making breakthrough in GIC-related-synthesis.

We thank the reviewer for this very positive evaluation of our work.

However, several parts should be revised as follows.

C1. The authors have mainly used K⁺ as an intercalant and PhCN for a solvent, and I wonder why the authors have chosen those materials among various candidates. To generalize the reaction that is so-called "solvent-reduction-induced and electrostatically driven mass transport", synthesis with other various intercalants and solvents may be done. For example, K⁺ can be replaced to Rb⁺, Cs⁺, or [Na-ether]⁺ and PhCN might be changed to other electron-trapping solvents.

We used predominantly potassium intercalated graphites, because these are the most frequently investigated and easy to handle GICs. Our intention was to provide a proof of concept which does not exclude further systematic studies employing other metal intercalants. It is reasonable to assume that other alkaline metals will behave similarly. However, in order to respond to the point addressed by the reviewer at this stage, we also carried out experiments with Rubidium in the meantime. Therefore, we produced a stage I intercalation compound with Rubidium (RbC₈). The titration with benzonitrile led to the same quantitative electron transfer from GIC to PhCN as observed for the potassium GICs (Fig. 1).

The reviewer is absolutely right that it is to be expected that other liquids with a similar or even higher reduction potential as benzonitrile will behave analogously. An advantage of PhCN⁻ certainly is that it does not undergo any unwanted side-reactions. We are convinced that in order to provide the desired proof of concept, it is a very suitable model solvent. In the future, we will provide related studies with other solvents as well.

C2. In figure 2, the authors claim that K⁺ diffuses out of graphite galleries by showing Raman spectroscopy data. However, I would like to ask X-ray diffraction result of K⁺-intercalated graphite (KC₈) and KC₈ after mixing with PhCN. The additional XRD data will clarify the structure of graphite (before and after exposure to PhCN) and will allow quantitative analysis of how much K⁺ goes out of graphite (in other words, the initial and final stage number).

We fully agree with the reviewer that XRD is an independent measure for the intercalation or deintercalation. As a consequence, the determination of XRD patterns will greatly strengthen our statement which based on Raman spectroscopy. Therefore we performed the requested XRD measurements of the graphite used as starting material, the KC₈ performed under inert conditions and the benzonitrile treated KC₈. We were very pleased to provide independent proof for the complete recovery (quantitative reoxidation and removal of intercalated K⁺) of graphite after the treatment of the GIC with benzonitrile (Supplementary Fig. 2). The XRD patterns of both KC₈ and recovered graphite with the interlayer distances of 3.35 Å (graphite) and 5.35 Å (KC₈) are exactly matching the literature values. From the XRD data, we can also conclude that the entire K has reacted to KC₈, while after PhCN treatment a full recovery of the graphitic reflections is observed. Thus, the XRD fully supports the Raman data in every respect. Moreover, these findings imply that in future, *in situ* Raman spectroscopy can be used as a self-contained tool for distinguishing of intercalated from parent graphite.

C3. In page 9 line 7, "To proof" should be changed into "To prove".

We corrected the spelling.

C4. In figure 4, it was shown that KC8 treated with PhCN produces higher-quality graphene compared to the one which were not treated with PhCN. That is to say, "electron trapping from reduced graphite" stimulates the exfoliation of graphite even without any ultrasonication. How and why does it happen? Would you explain fundamental reaction mechanism in detail with supporting data?

The beneficial action of PhCN is not to promote the initial exfoliation step of individual graphenide sheets from GICs, but represents a quantitative and non-destructive reoxidation agent. The exfoliation of KC₈ in DMSO is due to the Coulombic repulsion of the graphenide sheets (see also reference 14, 15) and is not due to the interaction with PhCN. We hope that this statement clarifies the misunderstanding that appeared to the reviewer.

5. The coloring of reacting species seems to have novelty in battery applications. The conventional battery system estimates SOC (state of charge) by measuring voltage, but it makes errors for some materials. However, the suggested method is expected to provide a new way of measuring the amount of charge in graphite. Therefore, additional comments about this would provide broader aspects to the readers.

We thank the reviewer for this very constructive comment. We therefore have added the following sentences into the manuscript: *"We are aware of the fact that our discovery of the straightforward charge monitoring using PhCN⁺ as a reporter molecule could find beneficial applications in battery technology. The current method for the state of charge (SOC) determination by measuring voltage is frequently associated with errors for some materials."*

Reviewers' comments:

Reviewer #1 (Remarks to the Author):

The authors have answered all of the queries raised by the reviewers. The work is now acceptable for publication in Nature Communications, although I still do not find overly compelling reasons to publish in such a high impact journal. I defer to the Editor. There are probably superior routes to graphene. The reviewer answered referee 1 by stating that "In the center of the flake, the deposition of the graphene sheets led to an overlapping of two individual graphene sheets". Is this bilayer graphene?

Publication is recommended.

Reviewer #2 (Remarks to the Author):

The XRD data provided in the revision are useful and do help support the major point that the initial reaction provides relatively pure stage 1 C8K. There are a few follow-up comments here, that I'd like the authors to consider:

1. The hkl indices for the graphite are not correct (020) should be (100) and etc.
2. The x axis scale does not make sense (the numbers proceed 25, 30, 25, 40, 35, 50)

3. The "discharged graphite" does show some high stage reflections (surrounding the main peak), not just graphite. So there is likely some high stage GIC remaining.

The second main point in my review was the need to compare a water-oxidized, and then thermally-treated, sample to one oxidized with benzonitrile. If there is not much difference, then the former route might be simpler and preferred.

I think the response does not address this concern adequately. The authors re-emphasized that oxidation with water leads to functionalization. But the point was that thermolysis could remove those functional groups. The authors indicate that "subsequent annealing step is required which makes the process very inappropriate compared to the straightforward inert oxidation with benzonitrile" But I'm not sure that's reasonable, i.e. would a treatment with water following by heating be more "straightforward" (I'm thinking the relevant issues here are cost, waste, time, etc) than washing with an organic solvent under inert conditions? If not, the application of this chemistry is less evident. The authors really need to show that water/thermolysis treatment leads to different results using the characterizational tools they have.

Finally, on the point about the detailed description of the reaction mechanism. This was a minor point in the overall impact of the manuscript, but I remain unconvinced that XRD or Raman provide new details on the reaction mechanism. I think the detailed description I've copied below is not directly supported by the data and should be removed:

"This can be explained by an initial heterogeneous electron transfer reaction from the solid GIC to the liquid benzonitrile phase. As a consequence, electrostatic forces are build up between the liquid phase, enriched with proximate, negatively charged PhCN⁻ anions and the solid graphitic phase, enriched with positive charges (still intercalated K⁺ counterions). The subsequent migration of the K⁺ leads to energy minimization of the entire system which is indicated by the intermediate step iii) in Figure 2. "

Reviewer #3 (Remarks to the Author):

The authors sincerely responded to the reviewers' comments and the supporting experiments were very carefully done. Therefore, I would like to have it published in Nature Communications. Additionally, there is a trivial error in page 2, line 18: "Also this latter questions is directly associated..." needs to omit "s" in "questions".

Author response to reviewer comments

Reviewer #1 (Remarks to the Author):

The authors have answered all of the queries raised by the reviewers. The work is now acceptable for publication in Nature Communications, although I still do not find overly compelling reasons to publish in such a high impact journal. I defer to the Editor. There are probably superior routes to graphene. The reviewer answered referee 1 by stating that "In the center of the flake, the deposition of the graphene sheets led to an overlapping of two individual graphene sheets". Is this bilayer graphene?

We thank the referee for his kind evaluation and recommendation for publication of our work in Nature Communications. The referee is right stating that in the small area of overlapping flakes a small amount of bilayer nature can be determined by both AFM as well as SRM analysis. The amount of bilayer graphene can be quantified by the G-mode intensity plot in Figure 4c. From there it can be seen that the amount of bilayer (yellow area) is less than 15%.

Reviewer #2 (Remarks to the Author):

The XRD data provided in the revision are useful and do help support the major point that the initial reaction provides relatively pure stage 1 C8K. There are a few follow-up comments here, that I'd like the authors to consider:

1. The hkl indices for the graphite are not correct (020) should be (100) and etc.

The referee is right that our notation of the reflections according to Kukesh *et al.* is commonly not used anymore and replaced the hkl indices as requested. We also updated the respective supplementary reference by the assignment of Wyckoff *et al.*

(<http://rruff.geo.arizona.edu/AMS/minerals/Graphite>)

2. The x axis scale does not make sense (the numbers proceed 25, 30, 25, 40, 35, 50)

We thank the referee for the careful reading. We replaced the transposed digits in the scale by the corrected values. The total scale was already correct.

3. The "discharged graphite" does show some high stage reflections (surrounding the main peak), not just graphite. So there is likely some high stage GIC remaining.

We fully agree with the referee that for the discharged graphite, the 002 reflection is slightly broadened compared to pristine graphite. However, the XRD pattern clearly proves the complete oxidation of KC_8 to graphite, since these reflections are completely vanished. Since the GIC was discharged in solution, the broadening of the 002 reflection can be easily explained in a way that reaggregation can lead to a slight disorder. This disorder is very small and therefore almost

negligible, as the Raman 2D mode also indicates intact AB stacking after PhCN treatment. The observed slight line broadening does not change the general picture of graphite reformation.

The second main point in my review was the need to compare a water-oxidized, and then thermally-treated, sample to one oxidized with benzonitrile. If there is not much difference, then the former route might be simpler and preferred.

I think the response does not address this concern adequately. The authors re-emphasized that oxidation with water leads to functionalization. But the point was that thermolysis could remove those functional groups. The authors indicate that "subsequent annealing step is required which makes the process very inappropriate compared to the straightforward inert oxidation with benzonitrile" But I'm not sure that's reasonable, i.e. would a treatment with water following by heating be more "straightforward" (I'm thinking the relevant issues here are cost, waste, time, etc) than washing with an organic solvent under inert conditions? If not, the application of this chemistry is less evident. The authors really need to show that water/thermolysis treatment leads to different results using the characterizational tools they have.

The referee is right that the covalent defect formation as a result of the water treatment can be reversed by an additional subsequent annealing step. The advantage in our approach however is that this energy and time demanding procedure is not necessary. Moreover the question of which method is superior for a possible graphene formation is not the key point and only a side aspect of our manuscript.

Finally, on the point about the detailed description of the reaction mechanism. This was a minor point in the overall impact of the manuscript, but I remain unconvinced that XRD or Raman provide new details on the reaction mechanism. I think the detailed description I've copied below is not directly supported by the data and should be removed: "This can be explained by an initial heterogeneous electron transfer reaction from the solid GIC to the liquid benzonitrile phase. As a consequence, electrostatic forces are build up between the liquid phase, enriched with proximate, negatively charged PhCN - anions and the solid graphitic phase, enriched with positive charges (still intercalated K⁺ counterions). The subsequent migration of the K⁺ leads to energy minimization of the entire system which is indicated by the intermediate step iii) in Figure 2. "

We thank the referee for this justified comment on the mechanism we suggested on page 7. We agree that the definite proof can be provided for the initial electron transfer from the GIC to PhCN and the reformation of graphite (this is valid for any postulation of a reaction mechanism). With respect to the individual steps in between, we provided an explanation which based on the available experimental information in our eyes is most reasonable. To address the remark of the referee, we rephrased our statement by "As a consequence for a possible mechanism, it can be assumed that electrostatic forces are build up between the liquid phase, ...".

Reviewer #3 (Remarks to the Author):

The authors sincerely responded to the reviewers' comments and the supporting experiments were very carefully done. Therefore, I would like to have it published in Nature Communications. Additionally, there is a trivial error in page 2, line 18: "Also this latter questions is directly associated..." needs to omit "s" in "questions".

We thank the referee for the careful reading of our manuscript and the very positive evaluation of our work. We removed the wrong "s" in the corresponding paragraph.